# Relative Validity and Reproducibility of a Semi-Quantitative Web-Based Food Frequency Questionnaire for Swiss Adults

**DOI:** 10.3390/nu17091555

**Published:** 2025-04-30

**Authors:** Sarah T. Pannen, Elsa Chevillard, Angeline Chatelan, Pedro Marques-Vidal, Silvia Stringhini, Robert Vorburger, Sabine Rohrmann, Nina Steinemann, Janice Sych

**Affiliations:** 1Division of Chronic Disease Epidemiology, Epidemiology, Biostatistics and Prevention Institute, University of Zurich, Hirschengraben 84, 8001 Zurich, Switzerlandnina.steinemann@uzh.ch (N.S.); 2Swiss School of Public Health, 8001 Zurich, Switzerland; 3Department of Nutrition and Dietetics, Geneva School of Health Sciences, HES-SO University of Applied Sciences and Arts Western Switzerland, Rue des Caroubiers 25, 1227 Carouge, Switzerland; elsa.chevillard@hesge.ch (E.C.); angeline.chatelan@hesge.ch (A.C.); 4Department of Medicine, Internal Medicine, Lausanne University Hospital, University of Lausanne, Rue du Bugnon 46, 1011 Lausanne, Switzerland; pedro-manuel.marques-vidal@chuv.ch; 5Unit of Population Epidemiology, Division of Primary Care Medicine, Geneva University Hospitals, Rue Gabrielle-Perret-Gentil 4, 1205 Geneva, Switzerland; silvia.stringhini@hug.ch; 6Faculty of Medicine, University of Geneva, 1205 Geneva, Switzerland; 7School of Population and Public Health, Edwin S.H. Leong Centre for Healthy Aging, Faculty of Medicine, University of British Columbia, 2206 E Mall, Vancouver, BC V6T 1Z3, Canada; 8Institute of Computational Life Sciences, School of Life Sciences and Facility Management, ZHAW Zurich University of Applied Sciences, Schloss 1, 8820 Wädenswil, Switzerland; voru@zhaw.ch; 9Institute of Food and Beverage Innovation, School of Life Sciences and Facility Management, ZHAW Zurich University of Applied Sciences, Grüentalstrasse 14, 8820 Wädenswil, Switzerland; sych@zhaw.ch

**Keywords:** dietary assessment, food frequency questionnaire, relative validity, reproducibility, usability, web-based FFQ

## Abstract

**Background/Objectives**: Food frequency questionnaires (FFQs) are widely used in large epidemiological studies to assess diet and elucidate its impacts on health. However, they must be validated in the target population before use. **Methods**: We assessed the relative validity, reproducibility, and usability of the Swiss eFFQ, a web-based, 83-item food frequency questionnaire, using a convenience sample of 177 adults (53.1% females, aged 18–75 years) from German- and French-speaking regions of Switzerland. The participants completed the Swiss eFFQ twice and kept a 4-day estimated food record (4-d FR). The dietary data were compared for energy, nutrient, and food group intakes by calculating mean group-level bias, performing the Wilcoxon signed-rank test, quartile cross-classification, weighted Cohen’s kappa (K_w_), and correlation coefficients. **Results**: The Swiss eFFQ was highly rated by the participants, with a completion time under 35 min, although it tended to underestimate nutrient and food intake compared to the 4-d FR. For 31 of 36 nutrients, fewer than 10% of the participants were classified in opposite quartiles. The median proportion of subjects classified in the same or adjacent quartile was 74.7% (median K_w_: 0.25). The median crude and de-attenuated Spearman correlation coefficients were 0.37 and 0.42 for nutrients and 0.45 and 0.52 for food groups, respectively. The median Spearman and intraclass correlation coefficients for the reproducibility of the Swiss eFFQ were 0.70 and 0.69 for nutrients and 0.70 and 0.61 for food groups, respectively. **Conclusions**: The Swiss eFFQ was shown to be reproducible and user-friendly, with acceptable accuracy in categorizing study participants based on food intake, and offers several advantages for dietary assessment of Swiss adult populations.

## 1. Introduction

Diet is a major risk factor for chronic disease, and the increasing prevalence of noncommunicable diseases, both globally and in Switzerland, underscores the urgent need to understand the relationship between dietary intake and chronic disease risk [1,2]. Research efforts exploring how diet influences health and disease outcomes relies heavily on the accurate assessment of dietary intake data [3,4]. This is traditionally achieved using food frequency questionnaires (FFQs), 24 h dietary recalls, or food records (FRs), with the choice of method depending on the research question and study design [5].

Within the field of nutritional epidemiology, FFQs are a valuable dietary assessment tool, commonly used to rank individuals within a population according to their dietary intake [6,7]. This method relies on the respondents indicating the frequency with which they consume items from a predefined list of food items. Compared with other dietary assessment methods such as FRs, FFQs impose a lower burden on respondents, are cost-effective, and allow for standardized data processing [7,8]. Moreover, the development of digital FFQs has further enhanced these advantages, enabling self-administration among large populations, the inclusion of adaptive follow-up questions based on previous responses, automated real-time data processing, and easier management of incomplete responses [9]. However, one inherent challenge of both traditional and digital FFQs is designing a food list that is both comprehensive and concise [6]. The accuracy of semi-quantitative FFQs is further limited by assumptions regarding food composition data, predefined portion sizes, and the respondents’ ability to accurately recall their food intake, alongside the cognitive demands of dietary self-assessment [3,10]. Therefore, before a newly developed FFQ can be applied in epidemiological studies, it is essential to assess its relative validity and reproducibility for the target population [11].

Differing from FFQs, FRs provide a more quantitative assessment of dietary intake by requiring individuals to record all foods and beverages consumed over several days, including details about each consumption events. This greater level of details accounts for their wide use as a reference method for validating FFQs, as was done in the present study [12]. However, FRs are more susceptible to reactivity and recording biases, such as changes in diet caused by socially desirable behaviors [7,13], and they are unsuitable for large-scale surveys due to their higher cost and participant burden [14,15].

Switzerland presents a unique context for nutritional epidemiology, shaped by its linguistic and cultural diversity [16,17,18,19], as well as regional variations in dietary habits [20,21] and mortality [18]. However, the limited number of FFQs available for use in the Swiss population are mostly outdated or have only been validated in one of the national languages [22,23,24]. To address these limitations and support standardized dietary assessments in Switzerland, we developed an electronic FFQ called the “Swiss eFFQ,” with the data-driven development process described in detail elsewhere [25].

The aim of this study was: (i) to assess the relative validity, reproducibility, and usability of the Swiss eFFQ in a German- and French-speaking Swiss study sample; and (ii) to conduct a comprehensive analysis of the results to identify the strengths and limitations of the newly developed eFFQ and facilitate its optimal application in epidemiological studies.

## 2. Materials and Methods

### 2.1. Study Population and Design

The online validation study was conducted from January to August 2023 at two study centers in Switzerland (located in the cantons of Zurich and Geneva). Participants eligible for the study were between the ages of 18 and 75, could speak German or French, had been residents in Switzerland for more than one year, and had regular access to the Internet. Additionally, participants were eligible if they (1) had not experienced a significant change in their body weight (≥10%) or made significant changes to their dietary habits in the three months prior to the study start, (2) were willing to maintain their dietary habits, (3) did not plan to participate in an interventional study for the duration of the study, and (4) were not pregnant or breastfeeding. Participants were recruited through various channels, including e-mail, social media, flyers, poster campaigns, and word of mouth. Eligible individuals were selectively invited to participate in the study, with the aim of reaching a balanced distribution by sex and age within the study population.

After signing an electronic informed consent, participants were asked to complete initial online questionnaires covering demographics and anthropometrics, the Swiss eFFQ (first time) and a corresponding usability questionnaire (Phase 1, T0). The usability questionnaire included ten different aspects of usability related to the Swiss eFFQ (see Appendix A), and participants were asked to select one of the following responses on a 7-item Likert scale: strongly disagree, disagree, somewhat disagree, neither agree nor disagree, somewhat agree, agree, or strongly agree. After baseline, participants were asked to complete a paper-based 4-d FR at two to four weeks (Phase 2, T1) and the Swiss eFFQ for a second time at three months (Phase 3, T2). An overview of the study procedure is shown in Figure 1. Participants received financial compensation in the form of grocery store gift cards (at T1 and after T2) and, if interested, a written evaluation of their diet at the end of the study. The Ethics Committee of the Canton of Zurich reviewed the research project summary and concluded that no formal ethical approval was required.

Of the 183 participants who were officially enrolled in the study and completed Phase 1, 176 (96.2%) and 170 (92.9%) participants completed Phase 2 and Phase 3 of the study, respectively. An overview of the study participant flow is shown in Appendix A, and further information on data collection and management is provided in the Appendix A.

### 2.2. Methods of Dietary Assessment

#### 2.2.1. Swiss eFFQ

The Swiss eFFQ, a web-based food frequency questionnaire (available in German and French), was used to collect food consumption frequency data. As reported earlier, its characteristics and food list were developed based on the data from the first Swiss National Nutrition Survey menuCH (*n* = 2085 participants) [21,25]. The 83-item Swiss eFFQ has been optimized to assess dietary intake in the culturally diverse Swiss adult population and measured the frequency of consumption over the past four weeks using ten frequency categories ranging from “never” to “more than 5 times a day” [25]. Each question was accompanied by a food picture illustrating an average portion size. Using population-based standard portion sizes, the Swiss eFFQ web platform automatically calculated daily energy, nutrient, food, and food group intakes by linking to the Swiss (version 6.4, SFDB [26]) and the German (version 3.02, BLS: Bundeslebensmittelschlüssel [27]) food composition databases [25]. Over 75% of the food items used were sourced from the SFDB, with BLS data utilized only when a corresponding food item was unavailable in the SFDB. The detailed iterative development process for the new web-based Swiss eFFQ, its overall structure and features has been published [25].

#### 2.2.2. Four-Day Food Record (Reference Method)

Participants received a paper template for a 4-d FR along with a digital (and if preferred, a printed) food picture book containing 119 series of five to six portion size images and approximately 60 actual household measurements, as well as instructions for completing the FR [28]. They were advised to provide detailed information on the type, portion size, and methods of preparing the beverages, foods, and ingredients of the recipes consumed. In addition, they were asked to complete the estimated 4-d FRs on four consecutive days, starting either on Wednesday or Sunday, and to return it to the study center using a prepaid envelope. Upon receipt, the FRs were checked for completeness by nutrition- and/or dietetics-trained study staff. To derive nutrient and food group intakes from the 4-d FRs, the staff transcribed the handwritten 4-d FR details into PRODI^®^ Swiss (version 7.0, nutricompass, Hofstetten, Switzerland), a nutrient analysis software linked to the Swiss (version 6.4, SFDB [26]) and German (version 3.02, BLS: Bundeslebensmittelschlüssel [27]) food composition databases. SFDB data were prioritized whenever available, and BLS data were only used if the specific food item was unavailable in the SFDB.

### 2.3. Sample Size Calculation

For dietary assessment validation studies, a sample size of at least 50, and preferably between 100 and 200, is recommended [6,11]. For this study, the primary outcome was the agreement between dietary intakes estimated from the 4-d FR (reference method) and those obtained with the Swiss eFFQ. The outcome variables were continuous. To estimate the sample size for the Bland–Altman method [6,29], we set α = 0.05 (Type I error) and β = 0.20 (80% power). Assuming μ = ±0.1 (expected mean of differences between measurements by the 2 methods, ±100 kcal; unit: kcal/1000), σ2 = 1 (expected standard deviation of differences), δ = 2.7 (pre-defined clinical agreement limits or maximum allowed difference between methods) and well-behaved data, the required sample size was 66 [29].

### 2.4. Statistical Analysis

Energy, nutrient, and food group intakes at T0, T1, and T2 were evaluated for normal distribution using the Shapiro–Wilk test and Q-Q plots, showing a clear deviation from normality for most variables. Macronutrient and alcohol intakes were additionally expressed as a percentage of energy intake (E%), and residual energy-adjusted nutrient intakes were calculated using the residual method [30]. Descriptive statistics were used to characterize the study population and their dietary intakes, and results are presented as medians with 25th and 75th percentiles (interquartile range, IQR) and frequencies with percentages (*n* (%)). To assess the relative validity between the Swiss eFFQ- and 4-d FR based on energy, nutrient, and food group intakes (T0 vs. T1), the following statistical analyses were performed. To compare differences in estimated intakes at the group level, the mean group-level bias (mean intake FFQ1 (T0))/(mean intake 4-d FR (T1)) × 100 − 100) and the Wilcoxon signed-rank test were calculated [31,32,33], where group-level biases smaller than 10% were considered good [31,32]. To assess the ability of the Swiss eFFQ to rank individuals based on their dietary intake in comparison to the 4-d FR, intake estimates were categorized by quartiles. The analysis then involved calculating the proportion of participants accurately classified within the same, same or adjacent quartiles, as well as those grossly misclassified, i.e., those classified highest by the Swiss eFFQ and lowest by the 4-d FR, or the reverse [31,34]. A threshold of ≤10% of participants misclassified into the opposite quartile was considered as good agreement at the individual level [31]. In addition, the weighted Cohen’s kappa statistic (K_w_) was calculated based on a 3 × 3 frequency table using the weightings assigned as 1 for complete agreement, 0.5 for partial agreement and 0 for complete disagreement [35]. The K_w_ values were interpreted as follows: over 0.80 as very good, 0.61–0.80 as good, 0.41–0.60 as moderate, 0.21–0.40 as fair, and less than 0.20 as poor agreement, as outlined by Masson et al. [35]. Bland–Altman analysis was performed to calculate limits of agreement for all dietary intake parameters, and plots were used to examine the differences and biases for energy and macronutrient E% values between the two dietary assessment methods [36,37]. The strength and direction of associations between estimated intakes using the two dietary assessment methods were evaluated using crude, de-attenuated, and de-attenuated residual energy-adjusted Spearman’s correlation coefficients (SCCs; due to the presence of tied values, Spearman’s *p*-value approximation was used to assess statistical significance). The SCC values were interpreted as follows: a result of ≥0.50, 0.20–0.49, and <0.20 indicated good, acceptable, and poor association, respectively [31]. Ratios of within- to between-person variation (λ = S_w_^2^/S_b_^2^) were calculated from the 4-d FRs using one-way ANOVA. De-attenuated SCCs were then calculated using the formula r_c_ = r_o_ √(1 + λ/k), where r_c_ is the de-attenuated correlation coefficient, r_o_ is the crude (observed) correlation coefficient, λ is the aforementioned ratio, and k is the number of replicates available for the reference method (in this case, 4 days) [23,38,39,40]. To assess the reproducibility (test-retest reliability) of the Swiss eFFQ over time, the dietary intakes derived from the first Swiss eFFQ (T0) were compared with those from the second application of the Swiss eFFQ (T2). The statistical methods used for this were cross-classification analysis, mean group-level bias, Lin’s concordance correlation coefficient (LCC; reproducibility index that evaluates the agreement between two measures) [22,41], intraclass correlation coefficient (ICC; high value indicates a low within-person variation) [42], and SCC. All statistical analyses were performed with R software (version 4.3.1 for Windows, R Foundation for Statistical Computing, Vienna, Austria).

## 3. Results

### 3.1. Baseline Characteristics of the Study Population

The participants who completed at least two of the three dietary assessments (*n* = 177, 53.1% female) were included in the analyses. The median age was 48.0 (IQR 31.0, 62.0) years, and the median body mass index (BMI, kg/m^2^) was 22.8 (IQR 21.1, 25.9). Most participants led a healthy lifestyle (Table 1). The characteristics of the study participants were mostly similar in both study centers, but the proportion of people following a vegetarian or vegan diet was higher in the German-speaking (18.2%) study center compared to the French-speaking one (3.8%).

### 3.2. Relative Validity

The study’s assessment of the Swiss eFFQ’s relative validity involved comparing estimated daily dietary intakes from the Swiss eFFQ at T0 with the mean of the 4-day FRs at T1 (*n* = 176; Table 2 and Table 3, Figure 2, Figure 3 and Figure 4, and Appendix A). The median estimated energy intakes were 1638 kcal/d by the Swiss eFFQ and 2023 kcal/d by the 4-d FR (*p* < 0.001). For most nutrients (including energy, hereafter referred to as nutrients), the Swiss eFFQ provided lower intake estimates than the 4-d FR, and 9 of 36 nutrients had a group-level bias <10% (Table 2 and Appendix A). With a median group-level bias of −19.8%, the values ranged from −36.1% for vitamin A activity (retinol equivalents, RE) to 6.0% for fat E% (Table 2 and Appendix A). A median of 35.8% of participants were correctly classified in the same quartile of nutrient intakes using both dietary assessment methods, ranging from 30.1% (fatty acids, polyunsaturated) to 51.1% (alcohol). Good agreement at the individual level (≤10% of participants classified in the opposite quartile) was observed for 31 of the 36 nutrients (Table 2 and Appendix A). Of the 21 food groups, the Swiss eFFQ overestimated the intake of one food group (non-alcoholic beverages) and underestimated the intake of nine food groups (*p* < 0.05), with group-level biases ranging from −46.4% (salty snacks) to 10.8% (fruits and fruit products) and a median of −23.2% (Table 3).

The median K_w_ values were 0.25 for nutrients and 0.30 for food groups (Table 2, Table 3 and Appendix A). A poor agreement was observed for nine nutrients, whereas an acceptable agreement was observed for 27. Out of the 14 food groups for which it was possible to perform a K_w_ analysis (distributions that allowed for the creation of unique tertiles), four food groups had a poor and ten food groups had an acceptable agreement. Notably, none of the nutrient or food group parameters demonstrated a good agreement based on the K_w_ analysis.

The median crude SCC values were 0.37 for nutrients and 0.45 for food groups (Appendix A, green lines in Figure 2 and Figure 3). A good association was observed for three nutrients (alcohol, alcohol E%, dietary fiber E%) and seven food groups. Most nutrient and food group values had an acceptable strength of association between the Swiss eFFQ and the 4-d FR. The ratio of within- to between-person variance calculated from the 4-d FR had a median of 0.90 for nutrients and of 1.21 for food groups (Appendix A). De-attenuation moderately improved the SCCs (median of 0.42 for nutrients and 0.52 for food groups; purple line in Figure 2 and Figure 3) and the de-attenuated residual energy-adjusted SCCs were comparable to or slightly higher than the crude nutrient SCCs, with a median of 0.46 (yellow line in Figure 2 and Appendix A).

The Bland–Altman plots for energy and E% values of carbohydrate, protein, and fat intakes are shown in Figure 4. On average, the Swiss eFFQ values were lower than the 4-d FR values for energy, carbohydrates E%, and protein E%, but they were higher for fat E%. Although the mean bias appeared to be constant for all intake levels, there were larger absolute differences between the methods with higher mean energy intakes but not with higher macronutrient E% values. Overall, for 14 of 36 nutrients and 8 of 21 food groups analyzed, ≥95% of the study participants fell within the limits of agreement (Appendix A).

### 3.3. Reproducibility

The nutrient and food group intakes estimated by both Swiss eFFQ measurements are shown in Table 2, Table 3 and Appendix A (*n* = 170). Overall, the differences between the first and second Swiss eFFQ were small. The median group-level difference between the two Swiss eFFQ measurements was 4.9% for nutrients (ranging from −13.4% for alcohol E% to 23.2% for vitamin C) and 4.0% for food groups (ranging from −20.1% for salty snacks to 24.4% for legumes). 

Most nutrient intakes had moderate to good reproducibility with a median crude SCC of 0.70. The median ICC was 0.68 (Table 2 and Appendix A). For food groups, the reproducibility parameters had a median of 0.70 for the crude SCCs and 0.61 for both ICC and LCC. Bakery products, cakes, and pastries (SCC) and salty snacks (ICC and LCC) showed the lowest reproducibility, whereas alcoholic beverages (SCC) and vegetarian dairy product replacements (ICC and LCC) showed the highest reproducibility (Table 3). On average (median), 90.0% (nutrients) and 86.3% (food groups) of the participants were consistently categorized into the same or an adjacent quartile in both the first and second Swiss eFFQ measurements (Table 2, Table 3 and Appendix A).

### 3.4. Usability

The usability questionnaire was completed by 166 of the study participants (Figure 5). Over 89.1% of them rated the Swiss eFFQ as easy to answer and intuitive to use, and they indicated that the questions and instructions were clear and easy to understand. The food pictures provided in the Swiss eFFQ were not considered helpful by 20.5% of the participants. Almost all the participants stated that the user interface of the Swiss eFFQ was clearly structured, the visual design was appealing, and they would be willing to complete the Swiss eFFQ again (Figure 5). The median time to complete the first Swiss eFFQ was 34.0 min (IQR 24.0, 46.0, *n* = 177) and 25.5 min for the second (IQR 18.4, 39.9, *n* = 170).

## 4. Discussion

The present study assessed the relative validity, reproducibility, and usability of the web-based Swiss eFFQ, which was designed to assess dietary intakes among Swiss adults speaking German or French [25]. Overall, the semi-quantitative Swiss eFFQ had an acceptable ranking ability, with a satisfactory agreement between cross-classifications in the Swiss eFFQ and the 4-d FR for most nutrient and food group intakes. While the estimated absolute dietary intakes tended to be lower in the Swiss eFFQ compared to the reference measure, the strength of the association (SCC) between the two assessments was good or acceptable for energy, macronutrients (except for MUFAs), and the majority of food groups. The reproducibility of the Swiss eFFQ was indicated by SCC, ICC, and LCC values exceeding 0.50 [46] for all nutrients and food groups except for retinol, salty snacks, and savory sauces, indicating reasonable reliability. Finally, the usability aspects of the Swiss eFFQ were generally rated as high, and the median time to complete the 83-item Swiss eFFQ was less than 35 min.

Given that the participants were active (Table 1), the energy intake estimated by the eFFQ—1638 (1357, 2003) kcal/day—is likely to be an underestimation when compared to the estimate from the 4-d FR—2023 (1715, 2349) kcal/day. Under- and overestimated total energy intake is a common limitation of FFQs, depending on the completeness of the food list and the suggested portion sizes [3,6]. Consistent with our findings of an overall lower energy intake estimated by the Swiss eFFQ compared to the 4-d FR, most nutrients and food groups were underestimated and showed group-level biases higher than 10%. In contrast, several other studies have reported that FFQs overestimated intake [47], possibly due to differences in their characteristics (e.g., grouping and number of food items) [11]. For example, a meta-analysis revealed a tendency for FFQs to overestimate dietary intake, and while the median number of food items in these FFQs was 126, the range was quite broad, extending from 18 to 322 items [47]. It is probable that the Swiss eFFQ’s low number of food items (*n* = 83), along with the way these items were grouped and the application of standard portion sizes, may have contributed to the consistent underestimation of intake that we observed.

In terms of the relative validity of energy intake, our results revealed an acceptable group-level bias, with more than 95% of individuals (good) falling within the limits of agreement. Less than 10% of the participants were classified in the opposite quartile, indicating good cross-classification ability, but a low K_w_ value suggested a poor ranking ability. The strength of association for crude and de-attenuated SCCs was acceptable and comparable to the correlation coefficients observed in previous Swiss FFQ validation studies [22,23]. Therefore, the relative validity of energy can be considered acceptable, but the estimated absolute energy intake should be interpreted with caution.

When the energy intake was considered and the relative validity of macronutrient E% values was assessed, notable improvements in the association strength (SCC) and ranking ability (K_w_) were observed compared with crude intakes. De-attenuated SCCs derived from crude macronutrient intakes were found to be lower than the pooled effect estimates from a meta-analysis by Cui et al. [47]. Nevertheless, our estimates based on macronutrient E% values showed comparable strengths of association between the Swiss eFFQ and the 4-d FR for all macronutrients, closely aligning with the pooled effect estimates derived from crude values reported in the aforementioned meta-analysis [47]. Similarly, despite differences in the specific nutrients included, the overall medians of the crude (0.38), de-attenuated (0.54), and energy-adjusted (0.41) correlation coefficients from all nutrients reported in the meta-analysis [47] were comparable to our findings (crude: 0.37, de-attenuated: 0.42, de-attenuated residual energy-adjusted: 0.46). Compared to two previous (relative) validation studies of paper-based FFQs conducted in the German-speaking [23] and French-speaking [22] regions of Switzerland, our results indicated higher strengths of association (based on SCC values) for carbohydrates and fiber, but lower strengths for fat and protein.

Given that the primary aim of the Swiss eFFQ was to categorize individuals based on their dietary intake, the cross-classification analysis results are crucial in assessing its relative validity for potential future applications. Although we employed quartiles for cross-classification in our study, which differ from the tertiles used in the referenced FFQ-validation study conducted in the French-speaking region of Switzerland [22], our findings reveal that the proportion of individuals misclassified into the most extreme categories for energy and macronutrient intake was comparable to or lower than those reported in the cited study [22]. Furthermore, our cross-classification results were consistent with those reported in a scoping review on electronic FFQs by Zainuddin et al. [48]. With the proportion of participants classified in the same or adjacent nutrient quartile ranging from 52.0% to 92.3% [48], our results (67.1–92.6%) indicate a classification ability that is comparable to that of other electronic FFQs [48]. Although weighted kappa coefficients for nutrients were only reported in three of the studies reviewed, the values observed in our study fall within a comparable range and indicate an acceptable ranking capability of the Swiss eFFQ for the majority of nutrients [48]. Compared to the other macronutrients, the ranking ability of the Swiss eFFQ was more modest (K_w_ values indicating a poor result) for crude fat and MUFA intake [35]. As suggested by Molag et al., a possible explanation could be that FFQs with shorter food lists (<100 items) tend to perform more poorly at ranking participants than FFQs with more extensive food lists [8]. This issue was particularly noticeable for fat and protein, which derive from a variety of different food sources [8]. Although this pattern was not evident for protein in our study, this hypothesis might partially explain the poorer ranking performance of fat compared to the other macronutrients with the Swiss eFFQ. However, in addition to the possible explanation of bias due to foods not being included in the Swiss eFFQ food list, the relative validity of fat intake may also be influenced by measurement errors in the queried food groups. In fact, food groups such as savory sauces, fats, oils, and cream, and mixed dishes, which are relevant sources of fat and MUFA intake, showed low strengths of association and poor ranking abilities.

The possible explanations are multiple. Firstly, these three food groups consisted of a limited number of food items with large variations in portion sizes, leaving them more susceptible to influences by individual responses. For example, the amounts of sauces, oils, and fats added to dishes varied greatly between the individuals, whereas the Swiss eFFQ accounted for a unique standard portion size for all. Secondly, the errors in reporting fats, oils, and cream might be caused by a lack of awareness or difficulty in remembering, especially for participants who do not usually cook. Thirdly, occasionally-consumed food items such as salty snacks, mixed dishes, and savory sauces may have been inadequately represented by the limited duration of the 4-d FR (i.e., compromised measurement of day-to-day variations). Similar challenges with the relative validity of food groups such as mixed dishes, savory sauces [49], and fats, oils, and cream [49,50] have also been observed in other web-based FFQ validation studies. Consistent with the findings reported in the literature [23,49,50], the food groups that were consumed in larger quantities and on a daily basis, such as meat, dairy, grain and bakery products, have been shown to have higher correlations and ranking abilities, indicating a satisfactory relative validity of the Swiss eFFQ for assessing the intake of most food groups.

Overall, the Swiss eFFQ demonstrated a good reproducibility for estimating both nutrient and food group intakes. With SCCs ranging from 0.59 to 0.88 for nutrients and from 0.53 to 0.88 for food groups, the Swiss eFFQ showed good to superior reproducibility compared with the reliability standards for these dietary assessment tools (correlation coefficients 0.50–0.80) [6]. Consistent with the observations in other studies, the total dietary intake estimates tended to be higher during the first measurement of the Swiss eFFQ compared to the second measurement, which might be due to a learning effect, as hypothesized by others [49]. This hypothesis is supported by the fact that respondents required less time to complete the second FFQ than the first. Additionally, in the present study, the ICCs [46,48,51], LCCs [22], and the proportion of participants categorized into the same quartile [48] met or exceeded the reproducibility metrics reported in other reviews [46,48,51] and studies [22] investigating the reproducibility of FFQs.

To the best of our knowledge, this is the first web-based, semi-quantitative FFQ that allows for automated data processing in the Swiss German- and French-speaking populations [25]. The Swiss eFFQ provided reasonable rankings of the individuals for most nutrients and food groups based on their self-reported dietary intake. In the design of the Swiss eFFQ, our primary aim was to minimize the respondent burden while maximizing its validity [25]. A particular strength of the Swiss eFFQ is that, despite its relatively short length and the use of standard portion sizes, its relative validity and reproducibility, for most dietary intake estimates, is comparable to longer FFQs or those collecting detailed portion size information [46,47,48,52]. Indeed, the favorable acceptance of the Swiss eFFQ by the study participants was indicated by positive ratings of its usability and a drop-out rate of 7%, which falls within the expected range [53]. Finally, a robust statistical analysis was conducted on the study results, leading to comprehensive insights about the validity of the Swiss eFFQ.

Nevertheless, certain limitations need to be considered. Firstly, our study relied on a convenience sample of volunteers, which may lead to selection bias and limit the generalizability of our findings. Secondly, we did not use biomarkers, which are suitable for evaluating the validity as an objective comparison to the self-reported dietary assessment data [14,54]. Although FRs are considered an appropriate reference method in validation studies, potential sources of error and bias in the data collection and data processing may have limited the accuracy of the collected data and thus affected the observed relative validity of the Swiss eFFQ [14]. Finally, by adhering to the recommended sequence of completing the test method before the reference method [11], a limitation of our study is the timing mismatch between the Swiss eFFQ and the 4-day FR assessment periods; the former considers the four weeks prior to baseline, and the latter two to four weeks after baseline (Figure 1). As the study participants may have become more aware of their diet after completing the first Swiss eFFQ, it cannot be ruled out that the observed results for the relative validity and reproducibility may have been influenced by changes in dietary habits. 

## 5. Conclusions

The Swiss eFFQ was shown to be a user-friendly dietary assessment tool with good reproducibility for the estimation of nutrient and food group intakes in a French- and German-speaking study population of healthy adults. In comparison to a 4-d FR, the total dietary intakes were underestimated by the Swiss eFFQ, but it showed an acceptable ranking ability for most nutrients and food groups. Overall, the Swiss eFFQ is well-suited for classifying individuals within populations in nutritional epidemiological research, offering a reliable method for assessing dietary intakes among Swiss adults speaking German and French.

## Figures and Tables

**Figure 1 nutrients-17-01555-f001:**
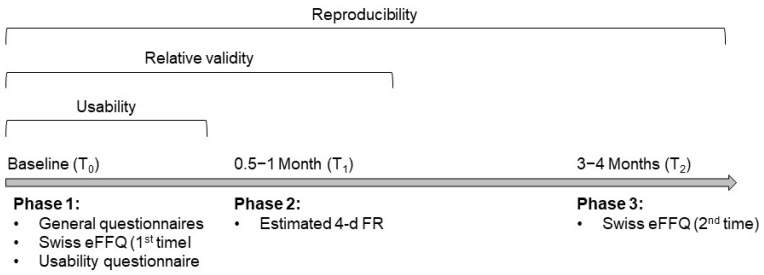
Overview of the study design, timeline, and measurement tools of the Swiss eFFQ validation study. Swiss eFFQ, web-based Swiss food frequency questionnaire; T, timepoint; 4-d FR, 4-day food record.

**Figure 2 nutrients-17-01555-f002:**
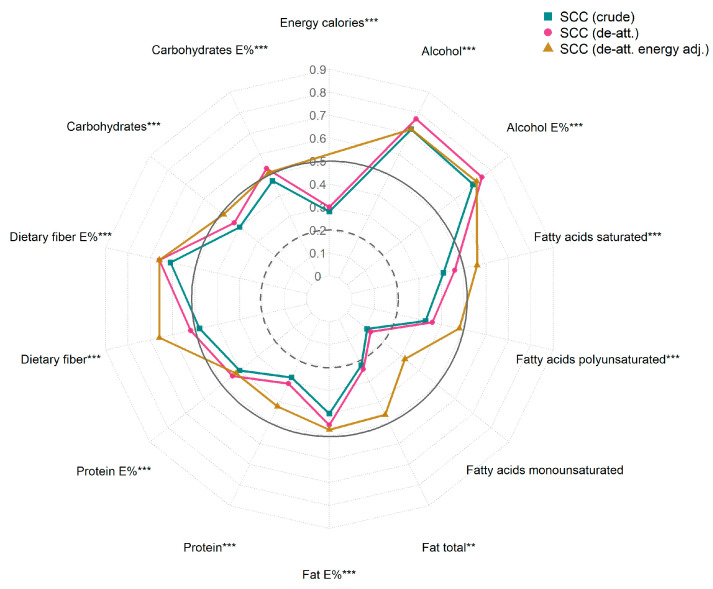
Associations between the Swiss eFFQ and 4-d food record estimated daily energy and macronutrient intakes (crude and E% values) using crude, de-attenuated, and de-attenuated residual energy-adjusted Spearman correlation coefficients. Adj., adjusted; de-att., de-attenuated; E%, percentage of total energy; SCC, Spearman’s correlation coefficient. Spearman’s *p*-value approximation was used to assess statistical significance of SCCs. Indicated statistical significance was calculated based on crude SCCs ** *p* < 0.01, *** *p* < 0.001). SCC values of ≥0.50, 0.20–0.49, and <0.20 were interpreted as good, acceptable, and poor association, respectively, and the corresponding cutoff values of 0.20 and 0.50 are indicated by the circles with the dashed and solid lines, respectively.

**Figure 3 nutrients-17-01555-f003:**
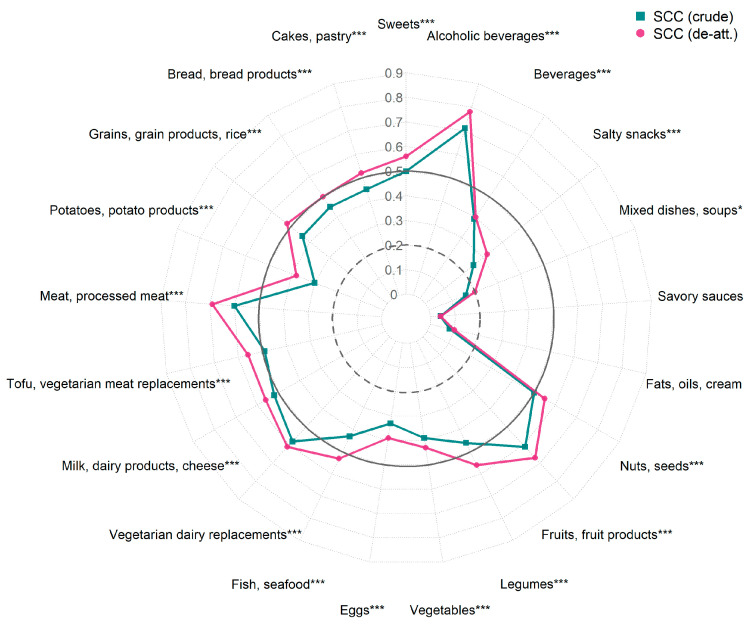
Associations between the Swiss eFFQ and 4-d food record estimated daily food group intakes using crude and de-attenuated Spearman correlation coefficients. De-att., de-attenuated; SCC, Spearman’s correlation coefficient. Spearman’s *p*-value approximation was used to assess statistical significance of SCCs. Indicated statistical significance was calculated based on crude SCCs (* *p* < 0.05, *** *p* < 0.001). SCC values of ≥0.50, 0.20–0.49, and <0.20 were interpreted as good, acceptable, and poor association, respectively, and the corresponding cutoff values of 0.20 and 0.50 are indicated by the circles with the dashed and solid lines, respectively.

**Figure 4 nutrients-17-01555-f004:**
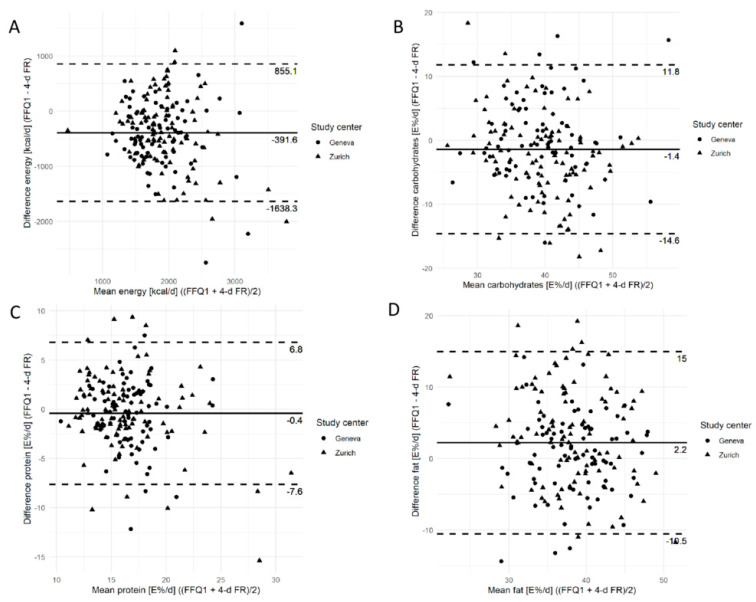
Bland–Altman plots of the differences in intake estimated using the first Swiss eFFQ and the 4-d food record, plotted against the mean of both methods stratified by study center for (**A**) energy, (**B**) carbohydrates (E%), (**C**) protein (E%), and (**D**) fat (E%). Horizontal lines represent the mean difference (solid black) and 95% limits of agreement (dotted lines). E%, percentage of total energy; FFQ1, first food frequency questionnaire at T0 (Phase 1); 4-d FR, 4-day food record.

**Figure 5 nutrients-17-01555-f005:**
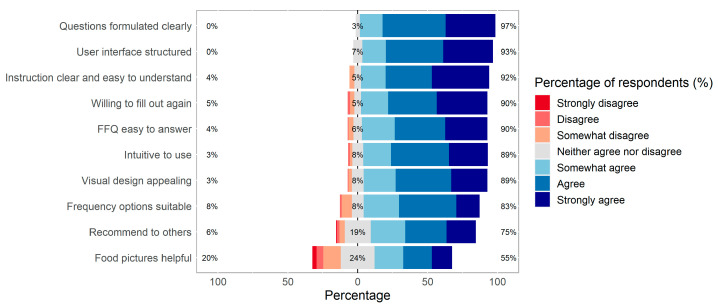
Usability of the Swiss eFFQ among 166 participants who completed the usability questionnaire in Phase 1. The y-axis represents responses by participants to statements referring to the Swiss eFFQ (see Appendix A), answered on a 7-item Likert scale (strongly disagree to strongly agree); the x-axis represents percentages of respondents’ responses.

**Table 1 nutrients-17-01555-t001:** Characteristics of the study participants who completed at least two dietary assessments (*n* = 177) included in the analysis of the relative validity and/or reproducibility of the Swiss eFFQ, overall and stratified by study center.

Characteristics	Overall	German-Speaking (Zurich Area)	French-Speaking (Geneva Area)
Number of participants (*n* (Sex female %))	177 (53.1)	99 (52.5)	78 (53.8)
Age [years] (median (IQR))	48 (31.0, 62.0)	48 (31.5, 62.5)	48 (31.0, 60.0)
Age category (*n* (%))			
18–37 years	59 (33.3)	34 (34.3)	25 (32.1)
38–57 years	58 (32.8)	30 (30.3)	28 (35.9)
58–75 years	60 (33.9)	35 (35.4)	25 (32.1)
BMI [kg/m^2^] (median (IQR)) *	22.8 (21.1, 25.9)	22.9 (21.5, 26.0)	22.6 (20.9, 25.8)
BMI category [kg/m^2^] (*n* (%))			
Underweight	9 (5.1)	6 (6.1)	3 (3.8)
Normal	112 (63.3)	62 (62.6)	50 (64.1)
Overweight	38 (21.5)	18 (18.2)	20 (25.6)
Obese	18 (10.2)	13 (13.1)	5 (6.4)
BMR [kcal/d] (median (IQR)) ^†^	1493 (1334, 1659)	1501 (1302, 1693)	1490 (1338, 1638)
PAL category (*n* (%)) ^‡^			
Low	12 (6.9)	9 (9.2)	3 (3.9)
Moderate	61 (34.9)	36 (36.7)	25 (32.5)
High	102 (58.3)	53 (54.1)	49 (63.6)
Smoking status (*n* (%))			
Never	99 (55.9)	57 (57.6)	42 (53.8)
Former	64 (36.2)	34 (34.3)	30 (38.5)
Current	14 (7.9)	8 (8.1)	6 (7.7)
Education level (*n* (%))			
Primary/no degree	3 (1.7)	1 (1.0)	2 (2.6)
Secondary	43 (24.3)	25 (25.3)	18 (23.1)
Tertiary	131 (74.0)	73 (73.7)	58 (74.4)
Health status (*n* (%))			
Very bad to medium	12 (6.8)	8 (8.1)	4 (5.1)
Good to very good	165 (93.2)	91 (91.9)	74 (94.9)
Enjoy cooking (*n* (%)) ^§^			
Do not agree to neutral	42 (23.7)	25 (25.3)	17 (21.8)
Agree to strongly agree	135 (76.3)	74 (74.7)	61 (78.2)
In charge of food shopping (*n* (%)) ^§^			
Do not agree to neutral	33 (18.6)	16 (16.2)	17 (21.8)
Agree to strongly agree	144 (81.4)	83 (83.8)	61 (78.2)
Follow vegetarian/vegan diet (*n* (%))	21 (11.9)	18 (18.2)	3 (3.8)

BMI, body mass index; BMR, basal metabolic rate; PAL, physical activity level. * BMI categorization: <18.5 = Underweight, 18.5–24.9 = Normal, 25–29.9 = Overweight, ≥30 = Obese. ^†^ BMR was calculated with the gender-specific Harris–Benedict equation [43]. ^‡^ The level of physical activity was assessed using the short version of the International Physical Activity Questionnaire (IPAQ) [44]. According to IPAQ, participants were classified as low, moderate, or high. IPAQ data were incomplete for two participants (one in each study center), for whom no PAL was calculated [45]. ^§^ Responses by participants to the following statements: [I enjoy cooking for others and for myself] and [I buy most or all of my food/groceries myself] using a 7-item Likert scale: strongly disagree, disagree, somewhat disagree, neither agree nor disagree, somewhat agree, agree, or strongly agree. Items 1–4 and items 5–7 were combined.

**Table 2 nutrients-17-01555-t002:** Comparison of estimated daily energy and macronutrient intakes (crude and E% values) of the first Swiss eFFQ with the 4-d food records (relative validity, *n* = 176) and with the second Swiss eFFQ (reproducibility, *n* = 170).

Dietary Intake Variable	FFQ1 Median (IQR)	Relative Validity (FFQ1 vs. Mean of 4-d FR)					Reproducibility (FFQ1 vs. FFQ2)						
		4-d FR Median (IQR)	%Group-Level Bias ^†^	% SQ/SOAQ ^‡^	% EQ ^‡^	K_w_ ^§^	FFQ2 Median (IQR)	% Group-Level Bias ^†^	SCC ^‖^	LCC	ICC	%SQ/SOAQ ^‡^	%EQ ^‡^
Energy (kcal/d)	1638 (1357, 2003)	2023 (1715, 2349)	−18.6 ***	31.2/73.3	9.1	0.18	1538 (1286, 1946)	4.9 ***	0.70 ***	0.72	0.73	57.6/91.8	1.8
Carbohydrates (g/d)	155.8 (126.8, 196.1)	196.9 (164.7, 244.0)	−20.8 ***	35.2/73.3	4.6	0.27	146.7 (121.3, 183.7)	6.1 **	0.70 ***	0.77	0.78	48.8/85.9	2.4
Carbohydrates (E%/d)	38.7 (34.3, 42.4)	40.4 (34.7, 45.1)	−3.5 **	39.2/78.4	3.4	0.35	37.5 (34.1, 42.1)	0.7	0.73 ***	0.76	0.77	54.7/92.3	0.6
Dietary fiber (g/d)	18.0 (14.2, 24.0)	24.5 (18.3, 30.5)	−24.2 ***	39.8/84.1	4.0	0.30	16.0 (13.3, 22.2)	7.6 ***	0.71 ***	0.70	0.71	52.4/90.6	1.8
Dietary fiber (E%/d)	2.2 (1.9, 2.6)	2.3 (1.9, 3.0)	−8.8 ***	43.8/86.4	2.3	0.38	2.1 (1.8, 2.5)	2.3	0.80 ***	0.79	0.80	53.5/93.5	0.6
Protein (g/d)	66.2 (53.0, 79.8)	83.0 (67.8, 100.5)	−21.0 ***	34.7/72.2	9.7	0.22	61.9 (50.1, 79.4)	3.5 **	0.73 ***	0.69	0.69	54.1/91.8	1.2
Protein (E%/d)	15.5 (14.2, 17.9)	16.3 (14.3, 18.1)	−2.5	36.9/77.3	6.8	0.25	15.8 (14.0, 17.8)	−0.6	0.79 ***	0.84	0.84	58.2/95.9	0.6
Fat (g/d)	73.6 (58.4, 87.8)	84.7 (66.4, 103.8)	−14.2 ***	31.2/72.2	10.2	0.16	67.6 (53.3, 82.2)	5.6 ***	0.69 ***	0.67	0.68	51.2/91.8	1.8
Fat (E%/d)	38.9 (36.0, 42.5)	37.1 (32.9, 41.3)	6.0 ***	39.2/77.3	6.2	0.22	39.1 (35.1, 42.9)	0.6	0.59 ***	0.64	0.64	44.7/81.8	3.5
Fatty acids monounsaturated (g/d)	28.4 (22.3, 35.0)	30.9 (23.4, 36.8)	−7.6 *	30.7/67.0	11.4	0.09	26.3 (20.8, 33.6)	4.0 *	0.67 ***	0.65	0.66	51.2/87.1	2.4
Fatty acids polyunsaturated (g/d)	11.5 (8.9, 14.8)	12.2 (9.5, 16.1)	−10.9 *	30.1/71.0	4.6	0.24	11.0 (8.3, 14.0)	6.0 **	0.70 ***	0.67	0.67	51.8/89.4	1.8
Fatty acids saturated (g/d)	25.6 (20.2, 32.2)	31.6 (23.8, 40.9)	−20.3 ***	39.2/73.9	5.1	0.26	23.6 (18.0, 30.6)	6.7 ***	0.70 ***	0.70	0.71	57.6/91.8	1.8
Alcohol (g/d)	5.4 (1.8, 12.4)	4.0 (0.0, 16.5)	−21.0	51.1/92.0	1.1	0.54	5.5 (1.9, 12.9)	−10.5 *	0.88 ***	0.81	0.82	72.3/97.7	0.0
Alcohol (E%/d)	2.3 (0.7, 4.7)	1.5 (0.0, 4.8)	0.4 *	45.5/92.6	1.1	0.52	2.4 (0.9, 5.4)	−13.4 ***	0.86 ***	0.79	0.80	65.3/96.5	0.0

EQ, extreme (opposite) quartile; E%, percentage of total energy; FFQ1, first food frequency questionnaire at T0 (Phase 1); FFQ2, second food frequency questionnaire at T_2_ (Phase 3); ICC, intraclass correlation coefficient; IQR, interquartile range; K_w_, weighted Cohen’s kappa; LCC, Lin’s concordance correlation coefficients; SCC, Spearman’s correlation coefficient; SOAQ, same or adjacent quartile; SQ, same quartile; 4-d FR, 4-day food record. E% values = macronutrient intake [g] × macronutrient constant [kcal/1 g macronutrient]/energy intake [kcal] × 100. * *p* < 0.05, ** *p* < 0.01, *** *p* < 0.001. ^†^ Relative validity: group-level bias = (mean intake FFQ1)/(mean intake 4-d FR) × 100 − 100); Reproducibility: group-level bias = (mean intake FFQ1)/(mean intake FFQ2) × 100 − 100); Wilcoxon signed-rank test assessed with FFQ1 and 4-d FR, and FFQ1 and FFQ2, respectively. ^‡^ Cross-classification analysis results are expressed as percentage of participants classified in the same/same or adjacent, and extreme quartile of nutrient distribution by FFQ1 and the 4-d FR (relative validity) and FFQ1 and FFQ2 (reproducibility). ^§^ Calculation of K_w_ is based on tertiles. **^‖^** Spearman’s *p*-value approximation was used to assess statistical significance of SCCs.

**Table 3 nutrients-17-01555-t003:** Comparison of estimated daily food group intakes of the first Swiss eFFQ with 4-d food records (relative validity, *n* = 176) and with the second Swiss eFFQ (reproducibility, *n* = 170).

		Relative Validity (FFQ1 vs. Mean of 4-d FR)					Reproducibility (FFQ1 vs. FFQ2)						
Dietary Intake Variable (g/d)	FFQ1 Median (IQR)	4-d FR Median (IQR)	%Group-Level Bias ^†^	%SQ/SOAQ ^‡^	% EQ ^‡^	K_w_ ^§^	FFQ2 Median (IQR)	%Group-Level Bias ^†^	SCC ^‖^	LCC	ICC	%SQ/SOAQ ^‡^	%EQ ^‡^
Bread and bread products	54.5 (34.8, 71.8)	79.0 (43.2, 110.8)	−21.6 ***	33.3/78.1	5.5	0.26	52.4 (30.8, 73.7)	−1.3	0.74 ***	0.74	0.75	55.7/87.4	3.3
Grains and grain products, rice	93.8 (64.9, 126.5)	111.5 (64.4, 174.1)	−24.2 ***	35.5/78.1	4.9	0.32	84.4 (58.1, 122.6)	5.2 *	0.66 ***	0.66	0.67	45.4/85.2	2.7
Potatoes and potato products	32.8 (20.4, 45.6)	37.5 (0.0, 79.5)	−32.1 **			0.16	30.2 (19.6, 44.8)	4.0	0.57 ***	0.55	0.56	43.2/83.6	4.9
Bakery products, cakes, and pastries	20.5 (12.6, 34.8)	25.8 (7.5, 63.1)	−38.3 ***	42.6/78.1	5.5	0.29	19.0 (11.9, 28.2)	19.1 ***	0.53 ***	0.75	0.76	43.7/82.0	4.9
Sweets, sugar, desserts, and ice cream	27.0 (16.8, 41.9)	28.1 (13.8, 50.6)	−20.5	43.7/80.9	4.9	0.31	24.4 (13.6, 41.2)	−1.6	0.70 ***	0.60	0.60	44.3/83.6	2.7
Meat, processed meat, and sausage	72.3 (37.8, 99.6)	59.2 (23.8, 110.6)	−5.5	43.7/81.4	2.7	0.41	67.7 (38.4, 101.7)	−3.5	0.86 ***	0.77	0.77	56.8/92.9	1.1
Tofu and other vegetarian meat replacements	3.2 (0.0, 19.5)	0.0 (0.0, 0.0)	−20.7				3.2 (0.0, 19.5)	−4.8	0.79 ***	0.73	0.73		
Milk, dairy products, and cheese	123.9 (55.4, 194.6)	159.7 (81.1, 252.5)	−25.5 ***	38.3/80.3	2.7	0.35	102.6 (49.4, 192.7)	8.7 *	0.76 ***	0.51	0.51	50.3/90.2	1.6
Vegetarian dairy product replacements	0.0 (0.0, 9.7)	0.0 (0.0, 10.0)	−27.0				0.0 (0.0, 13.8)	−0.8	0.76 ***	0.82	0.82		
Fish and seafood	11.8 (9.3, 24.9)	8.5 (0.0, 37.8)	−33.9				11.8 (9.3, 24.9)	−2.8	0.79 ***	0.60	0.61	56.8/86.9	2.7
Eggs	11.1 (4.6, 11.1)	13.0 (0.0, 28.1)	−25.0				11.1 (4.6, 11.1)	18.1 **	0.64 ***	0.61	0.61	54.6/74.9	4.9
Vegetables	125.1 (93.5, 173.2)	188.5 (113.7, 274.1)	−34.4 ***	34.4/74.3	3.8	0.21	140.3 (91.6, 172.8)	−6.3	0.72 ***	0.60	0.61	48.1/85.8	1.6
Legumes	7.1 (2.9, 17.1)	0.0 (0.0, 25.0)	−21.0				7.1 (2.9, 17.1)	24.4 **	0.74 ***	0.69	0.70	56.3/89.6	0.6
Fruits and fruit products	207.3 (110.8, 325.5)	188.9 (99.3, 290.5)	10.8	42.1/85.2	2.7	0.45	153.1 (89.7, 260.0)	23.3 ***	0.68 ***	0.57	0.59	47.0/87.4	3.3
Nuts and seeds	3.5 (1.4, 12.7)	6.0 (0.0, 16.2)	−30.6 *			0.39	3.5 (1.4, 11.6)	13.5	0.66 ***	0.69	0.69	50.8/85.8	3.8
Fats, oils, and cream	18.8 (11.9, 25.2)	19.2 (10.7, 32.8)	−15.8 *	26.2/64.5	9.8	0.04	17.4 (10.6, 24.5)	10.2 *	0.65 ***	0.57	0.58	50.3/83.6	3.3
Savory sauces	15.9 (7.6, 21.9)	12.5 (5.0, 23.8)	−23.2	24.0/61.2	12.0	0.04	13.9 (6.5, 19.9)	7.4 *	0.65 ***	0.44	0.44	44.3/88.0	2.7
Mixed dishes and soups	82.8 (54.4, 117.6)	75.0 (2.2, 150.0)	−3.4			0.17	61.8 (40.5, 100.3)	21.8 ***	0.64 ***	0.57	0.60	44.3/85.8	2.2
Salty snacks	3.1 (1.2, 3.1)	0.0 (0.0, 10.0)	−46.4 *				3.1 (1.2, 3.1)	−20.1	0.59 ***	0.33	0.33		
Non-alcoholic beverages	1961.8 (1407.4, 2267.0)	1731.6 (1249.1, 2213.9)	5.6 *	35.5/77.6	7.1	0.33	1935.7 (1619.0, 2299.0)	−0.3	0.74 ***	0.75	0.75	55.2/89.1	2.2
Alcoholic beverages	75.0 (17.9, 166.4)	50.0 (0.0, 189.4)	−18.8				69.1 (17.9, 182.9)	−12.9 **	0.88 ***	0.79	0.79	55.2/92.9	1.6

EQ, extreme (opposite) quartile; FFQ1, first food frequency questionnaire at T0 (Phase 1); FFQ2, second food frequency questionnaire at T2 (Phase 3); ICC, Intraclass correlation coefficient; IQR, interquartile range; K_w_, weighted Cohen’s kappa; LCC, Lin’s concordance correlation coefficient; SCC, Spearman’s correlation coefficient; SOAQ, same or adjacent quartile; SQ, same quartile; 4-d FR, 4-day food record. * *p* < 0.05, ** *p* < 0.01, *** *p* < 0.001. ^†^ Relative validity: group-level bias = (mean intake FFQ1)/(mean intake 4-d FR) × 100 − 100); Reproducibility: group-level bias = (mean intake FFQ1)/(mean intake FFQ2) × 100 − 100); Wilcoxon signed-rank test assessed with FFQ1 and 4-d FR, and FFQ1 and FFQ2, respectively. ^‡^ Cross-classification analysis results are expressed as percentage of participants classified in the same/same or adjacent, and extreme quartile of food group distribution by FFQ1 and the 4-d FR (relative validity) and FFQ1 and FFQ2 (reproducibility). Values are missing (.) if distributions did not allow the creation of unique quartiles. ^§^ Calculation of K_w_ is based on tertiles. **^‖^** Spearman’s *p*-value approximation was used to assess statistical significance of SCCs.

## Data Availability

The original contributions presented in the study are included in the article/Appendix A, further inquiries can be directed to the corresponding author.

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
