# Peer review of "Relative Validity and Reproducibility of a Semi-Quantitative Web-Based Food Frequency Questionnaire for Swiss Adults"

_nutrients, 2025, doi:10.3390/nu17091555_

Round 1

Reviewer 1 Report

Comments and Suggestions for Authors

It's a very interesting study. However, there are a few aspects that the authors should improve upon.

  1. In the introduction, the authors should articulate the rationale for why readers must read the study and present its theoretical or practical contributions.
  2. The study presents a fairly complex set of findings. To ensure the clarity and accessibility of the findings for a broader audience, it would be advisable for the authors to distill them down to their essentials.
  3. As illustrated in Figure 2, the yellow line demonstrates a high level of uniqueness, while the other two lines exhibit significant similarity. The authors should clarify the differences and similarities between the lines.

Author Response

It's a very interesting study. However, there are a few aspects that the authors should improve upon.

  1. In the introduction, the authors should articulate the rationale for why readers must read the study and present its theoretical or practical contributions.

Response: Thank you for this comment which we carefully considered and implemented the following changes: We have revised the Introduction (lines 54-94) to better highlight the rational of the study and strengthened the aim to present more clearly the contribution of the study (lines 97-100).

  1. The study presents a fairly complex set of findings. To ensure the clarity and accessibility of the findings for a broader audience, it would be advisable for the authors to distill them down to their essentials.

Response: We agree that the results section is rather complex. We carefully went through the text and shortened it to make it more comprehensible. The number of changes is too large to be presented here.

  1. As illustrated in Figure 2, the yellow line demonstrates a high level of uniqueness, while the other two lines exhibit significant similarity. The authors should clarify the differences and similarities between the lines.

Response: Thank you. We now mention explicitly the different models in the text. Please see lines 312-324 in the Results section.

Reviewer 2 Report

Comments and Suggestions for Authors

The authors made an attempt to evaluate the relative validity of the Swiss eFFQ against a 4-day FR and to assess its reproducibility and usability in a German- and French-speaking Swiss study sample.

My comments are as follows:

Authors claim that Swiss eFFQ tended to underestimate nutrient and food intake compared to the 4-d FR. Why the 4-d FR was used as a reference? Maybe 4-d FR overestimated the intake of nutrient and food?

line 82 - Why do the authors consider important differences between national languages? In fact, no analysis with this respect was done in the study.

lines 11-112 - I assume that initial Swiss eFFQ and a second Swiss eFFQ were the same questionaires. Is that true, or some changes/amendment were made?

line 114 - Just for my curiosity -  were many participants interested in their diet evaluation?

point 2.4. Statistical analysis is presented in a descriptive manner, maybe it would be possible to present it, at least in part, in tabular of graphical form?

Author Response

The authors made an attempt to evaluate the relative validity of the Swiss eFFQ against a 4-day FR and to assess its reproducibility and usability in a German- and French-speaking Swiss study sample.

My comments are as follows:

  1. Authors claim that Swiss eFFQ tended to underestimate nutrient and food intake compared to the 4-d FR. Why the 4-d FR was used as a reference? Maybe 4-d FR overestimated the intake of nutrient and food?

Response: Thank you for your comment regarding the choice of the 4-d FR as a reference method in our validation study of the Swiss eFFQ. We fully agree that all dietary assessment methods have inherent limitations, and it is indeed important to critically consider the assumptions underpinning the use of any reference instrument. Four-d FR are a pragmatic and widely accepted reference instrument compared to FFQ (Willett 1998; Naska et al. 2017). Importantly, our objective was not to assert the absolute accuracy of either method, but rather to assess the relative validity of the eFFQ in estimating food and nutrient intakes.

 The statement that Swiss eFFQ tended to underestimate nutrient and food intake compared to the 4-d FR is supported by our energy intake estimates: the median (IQR) basal metabolic rate of our study population was 1493 (1334, 1659) kcal/day. Given that participants were classified as moderately to highly active (Table 1), the energy intake estimated by the eFFQ—1638 (1357, 2003) kcal/day—is likely to be an underestimation when compared to the estimate from the 4-d FR—2023 (1715, 2349) kcal/day. This discrepancy reinforces the well-documented tendency of FFQs to under or overestimate intake (Willett 1998; Naska et al. 2017).

We have clarified this in the text (lines 409-413):

“Given that participants were active (Table 1), the energy intake estimated by the eFFQ—1638 (1357, 2003) kcal/day—is likely to be an underestimation when compared to the estimate from the 4-d FR—2023 (1715, 2349) kcal/day. Under- and overestimated total energy intake is a common limitation of FFQ, depending on the completeness of food list and the suggested portion sizes (Willett 1998; Naska et al. 2017).

  1. line 82 - Why do the authors consider important differences between national languages? In fact, no analysis with this respect was done in the study.

Response: From previous studies, we know that differences in dietary habits exist in Switzerland (Chatelan et al. 2017). Therefore, differences in dietary habits between language regions were considered during the development of the questionnaire (a detailed description about this process has been published previously by our group [Pannen et al. 2023]). However, we decided not to conduct additional analyses on this aspect within the current study due to the already extensive scope of the paper. However, our aim is to explore the topic in more detail as part of a follow-up project.

  1. lines 11-112 - I assume that initial Swiss eFFQ and a second Swiss eFFQ were the same questionaires. Is that true, or some changes/amendment were made?

Response: Yes, both the initial Swiss eFFQ and the second Swiss eFFQ were identical; no changes or amendments were made between the two versions. We clarified this in lines117/118, 123/124, and 226/227.

Line 117/118: “participants were asked to complete initial online questionnaires covering demographics and anthropometrics, the Swiss eFFQ (first time) and a corresponding usability questionnaire”

Lines 123/124: “After baseline, participants were asked to complete a paper-based 4-d FR at two to four weeks (Phase 2, T1) and the Swiss eFFQ for a second time at three months (Phase 3, T2).”

Lines 226/227: “To assess the reproducibility (test-retest reliability) of the Swiss eFFQ over time, the dietary intakes derived from the first Swiss eFFQ (T0) were compared with those from the second application of the Swiss eFFQ (T2).”

  1. line 114 - Just for my curiosity -  were many participants interested in their diet evaluation?

Response: The vast majority of study participants indicated that they were interested in their diet evaluation. We included a specific question in the survey questionnaire asking whether participants wished to receive their individual dietary evaluation. Since this approach kept the barrier very low, around 98% of participants opted in, and we provided the evaluation results to them after the end of the study.

  1. point 2.4. Statistical analysis is presented in a descriptive manner, maybe it would be possible to present it, at least in part, in tabular of graphical form?

Response: We agree that the description of results is rather complex. According to reviewer 1, we shortened the text to make it more comprehensible. However, we believe that adding more figures will not improve comprehensibility. We rather keep the table to provide the opportunity for the readers to have a look at the results in detail.